# Empowering leadership: A conflict resolver and a performance booster for organizations

**Yi Wang** [ORCID] *

School of Management, Henan Institute of Economics and Trade, Zhengzhou, Henan, China

* wangyi920125@163.com

## Abstract

Organizational sustainability has become a critical challenge in the current era. This research purpose is to determine the impact of empowering leadership on conflict management and employees' performance for organizational sustainability. Furthermore, it also investigates the moderating impact of emotional stability on the relationship between empowering leadership, conflict management, and employee performance. Quantitative data for this research was collected from 512 middle-management-level employees from manufacturing firms in China. The partial least squares structural equation modelling results highlighted that empowering leadership positively impacts conflict management and employees' performance. Furthermore, the study showed that the organization's sustainability is possible with conflict management and employee performance when there is emotional stability. The theoretical grounding of this research closed a loop in the literature, and the findings are reliable for practice for organization sustainability.

**Data Availability Statement:** All relevant data are within the paper.

**Funding:** The author(s) received no specific funding for this work.

## 1. Introduction

The performance of any firm in a sustainable way is possible because of the performance of its employees. China is the largest economy in the world, and it has many public and private sector industrialization [1]. Chinese manufacturing sector firms face problems because of their diverse workforce and organizational performance. The Chinese public sector firms' management is interested in working for organizational sustainability, but middle management practices are hurdled to taking organizations sustainably [2]. Undoubtedly, conflicts between Chinese employees are increasing due to mismanagement in organizational culture. The problems related to employee conflicts based on organizational politics are leading organizations to decline in performance [3].Comparatively, it is necessary to improve conflict management to advance employees' performance for sustainable working. The manufacturing sector organizations in China have many workforces from different cultures based on their religious beliefs.

There is a need to improve the performance of Chinese manufacturing sector firms because the advancement in this performance will directly impact the sustainability of the market [4]. However, the performance of employees is a better determinant for checking the performance of any organization. Indeed, organizational politics also have a negative impact on the performance of employees [5]. When employees are not motivated to improve their performance

**Competing interests:** The author has declared that no competing interests exist

critically, they must work for better management. Although the purpose of any organization is to improve the performance of employees to advance sustainability, it is also determined that the increment in salary and other benefits only have an inconsistent impact on the performance of employees [6]. When the performance of any organization is reliable, the employees of such an organization are working in unity. The advancement of employees' working environments directly impacts their performance and leads them toward a better working direction [7].

Empowering leadership has become a critical factor in organisational work. It provides a sense of ownership for the employees, which boosts their performance. When employees feel comfortable about their work, they go for sustainable working to advance their behaviour in research progress. Indeed, empowering leadership motivates employees to work toward the long-term vision of the firm. In accordance, the management of conflict between the employees can be enhanced by empowering them. Moreover, when the employees have a sense of working freedom and autonomy, they have a productive approach to achieving their goals accordingly. Hence, the performance of the organisation is linked to boosting the employees' personalities with a sense of responsibility and efficient working.

The manufacturing sector organizations' performance is a practical challenge in China [8]. However, this practical problem is decreasing the economic performance of these organizations. However, the existing literature is reviewed critically to determine the possible factors that could help advance organizational sustainability. The previous studies showed that organizational sustainability can be advanced when employees are trained to work with proper motivation [9], better human resource department practices [10], and provide a competitive advantage to the employees. However, these studies have an open loop in the literature. The existing practical problems related to organizational sustainability in China have less theoretical knowledge. Therefore, existing literature has a proper gap in the body of knowledge that should be addressed on time. The reliability of employees' performance and their work with management support can improve organizational performance [11, 12]. With the conceptualization of this research, the following research has taken a step forward to find new factors that possibly impact organizational sustainability. This research is initiated to address the following questions.

Q1: *How does empowering leadership affect conflict management and employee performance in the manufacturing sector of China*?

Q2: *What is the impact of conflict management and employees' performance on organization sustainability*?

Q3: *What is the impact of emotional stability as a moderator between empowering leadership, conflict management and employees' performance*?

This study has significance because practical identification of factors that could contribute to conflict management, employee performance, and organization sustainability in China is required. Furthermore, this research aims to answer the above questions and close a loop existing in the literature. The following research is based on primary data collected from employees to generalise its findings. The scope of this study is limited to firms in the manufacturing sector in China. This research is divided into different subsections containing the review of existing literature, the methodology of the research, the statistical data analysis, the discussion of results considering the previous studies, and the final implications of the study. However, this research has some future directions for conducting further studies in this area of knowledge.

## 2. Review of literature

### 2.1 Transformational leadership theory

Transformational leadership theory refers to the leadership style that improves employees' productivity [13]. According to this theory, leaders influence employees for their productive output and behaviour [14]. The transformational leadership theory explains how leaders who empower their employees and effectively handle conflicts can have a significant impact on employee performance, potentially affecting the organization's long-term viability. Empowering leadership is a managerial approach that fosters a sense of ownership and motivation among employees, thereby encouraging proactive and innovative behaviour. This leads to improved performance outcomes. Furthermore, transformational leaders demonstrate exceptional conflict management skills by cultivating open communication and encouraging constructive conflict resolution. This method effectively reduces potential disruptions that could impede both productivity and team collaboration. Employee performance optimisation, achieved through the implementation of empowering leadership and effective conflict resolution strategies, is critical to achieving organisational sustainability. This is due primarily to its ability to reduce turnover rates, improve workplace culture, and improve adaptability to change. The presence of emotional stability has the potential to moderate the relationships mentioned above. Individuals who are emotionally stable, in particular, may benefit more from engaging in empowering leadership and conflict management practises. As a result, the positive effects on performance and sustainability are amplified [15].

### 2.2 Hypotheses development

When any organization's employees have different social values and working styles, they face conflicts. The conflicts of these employees are not suitable for their sustainable performance [16, 17]. The working style of employees matters a lot when their performance is required to take the firm to the top level [18]. The employees' leadership qualities motivate them to improve their influence over others to get the work done [19]. The middle management in any firm has a vital responsibility to work toward the goals of top management. Therefore, the employees must behave satisfactorily to improve their working performance. The leadership qualities of employees have a significant impact on their performance and working abilities [20]. However, when employees are motivated, they must have a way forward to work for the organization. Since any organization's performance is improved by its employees' performance, the management must trust their capabilities [21]. When the strategic management approach is used to empower the employees, this factor can influence the performance in the firm [22]. However, employees who are less motivated to improve their strategic performance can become more productive for sustainable working behaviour. Conflict management between the team becomes effective when there is appropriate employee support [23].

**H1: *There is a relationship between empowering leadership and conflict management.***

Every organization is working on a competitive style in China for its sustainability. The manufacturing sector business is required to improve the performance of employees for delivering standard products into the market [24]. The employees' skills help achieve sustainability in the organization's work. When the skills of any organization are set to meet the appropriate standards, the performance of the employees changes ([25]. The role of employees in any firm is to work for productive output, but the management should consider their ideas to improve the organizational working [26]. The empowering approach to the employees can facilitate their understanding of the organizational structure in a better way [27]. When the employees of any firm have a robust working approach and believe that the organizational support is

appropriate, they work better [28]. The sense of partial independence in thinking is the approach that can lead the employees into productive behavior [29]. The performance of employees increases and decreases over time, but strategic working helps to achieve sustainability in performance [30]. The workings of any organization can be improved when employees have the liberty to convey their innovative ideas to the management of these firms.

**H2: *There is a relationship between empowering leadership and employees' performance.***

The conflict management approach helps to improve the conflicts between the employees of any organization. Conflicts are expected when there is diversity in the workforce [31]. However, the employees are required to have appropriate behaviour that can support them in sustainable working [32]. The employees' innovative performance helps them get better opportunities when working in teams without any conflict [33]. The sustainability of the organization and the performance of employees can become strategic ways to improve organizational performance [34]. The management has a vital responsibility to reduce the conflicts between the employees and ensure their productivity [35]). Employees' access to their productive performance is a way forward approach to advancing their behaviour. Teamwork is required to improve the performance of any organization as this helps to achieve sustainability goals productively [35]. Many firms are motivating employees with an empowering leadership approach, but there is a need to improve their behaviour critically for a more productive approach [35]. When the management of any organization has the responsibility for empowering the employees, their performance increases to meet the organization's standards [36]. The employees' less productive attitude leads them to a negative way of working.

**H3: *There is a relationship between conflict management and organization sustainability.***

Modern-time businesses require the productive performance of their employees. These businesses are required to have employees in their hands for productive performance [37]. The available resources and business opportunities for employees can become a strategic way to increase productivity [38]. The sustainability of any organization is possible when there is a competitive advantage to the organization [39, 40]. Reliable working opportunities for employees can become a source of employee performance. Many firms are hiring talented employees to improve their performance [41]. However, teamwork is also required for the employees to advance their work productively. The availability of resources and training for the employees can lead to productive performance that is critical for their sustainable working [42]. No doubt, the performance of employees changes over time, but reliable opportunities for employees' performance can be possible when they have a strategic approach to advance their work [43]. The management is responsible for focusing on the resources and reducing the conflicts between the employees to improve their direction and advance the organizational performance [44]. When employees are motivated to perform in any condition, the chances of organizational sustainability are increased.

**H4: *There is a relationship between employees' performance and organization sustainability.***

Emotions are part of human life; people can't divorce their emotions. The productive performance of the employees is possible when they have opportunities to work for organizational sustainability [45]. The improved vision of employees for productive performance can provide them with opportunities to work for a competitive [16]. The employees become emotional when they observe different kinds of behaviour in the organization [46]. The management approach to empower the employees can help them become more productive because it helps them reduce their conflicts and work for the organization's sustainability [18]. When the

management of any organization has a vital focus on achieving organizational sustainability, the performance of these firms changes over time [47, 48]. However, the employees' psychological empowerment helps them overcome their issues to reduce their dependency on organizational performance [49]. The available resources for organizational performance and employees' productive approach can become a way forward to improve the employee's behaviour [50]. The advanced behaviour of the employees is possible when they have an approach to improving their productive behaviour in working [1]. Managing conflicts between employees is also a significant strategy to improve their behaviour for solid organizational competition in the market.

**H5: *There is a moderating role of emotional stability between empowering leadership and conflict management.***

The management has the responsibility to improve the performance of employees [51]. When the top management of any firm is working in a positive direction, and its strategic approach is to improve the performance of employees, there is a need to have a strong working relationship between the employees [52]. Every employee has a different set of values and working styles, but the organization's performance can become more productive when the leadership is empowered to improve organizational performance [53]. The strategic approach to improving organizational performance is necessary as management supports the employees to increase their productivity [54]. Many firms' management has different kinds of working approaches, but it is necessary to ensure the employees are working in the right direction. The sustainability working approach is possible when the empowering approach to the employees is implemented in the firm by its management [2]. The employees' innovative work leads them to resolve their conflicts and have a better approach to working [8]. Meanwhile, when the employees of any organization are not supported appropriately, there is a need to improve organizational performance and its advancements.

**H6: *There is a moderating role of emotional stability between empowering leadership and employees' performance.***

Fig 1 is shown below for better understanding of the framework of this study.

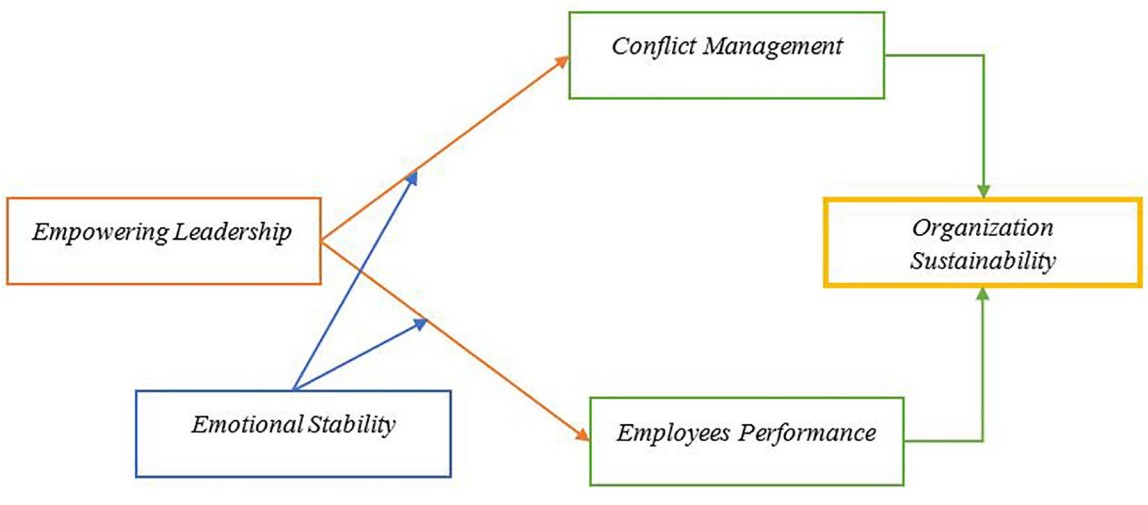

**Fig 1. Theoretical framework of study.**

## 3. Methodology

### 3.1 Sampling frame and technique

The research has considered the employees of manufacturing sector firms in China as the population of study. However, it is impossible to collect the data from every single population element. Therefore, this research has focused on the employees of manufacturing sector firms working in Wuhan, China. Wuhan city was selected for data collection because it is the largest city in Hubei and is considered a business hub in central China. The author randomly selected the 30 firms based on convenience and targeted the managerial employees for data collection. Furthermore, the random sampling approach was considered appropriate for collecting the data. When the sampling frame of any research is defined and the information about the target respondents is available, the random sampling approach is reliable [55].

### 3.2 Data collection and sample size

The data collection started from the start of June 2023 to the middle of July 2023. The total number of middle management level employees in sixty companies was 711. Seven hundred questionnaires were printed to collect the data, and each questionnaire was divided into different subsections containing the measurement for each variable. The survey was conducted physically for data collection. The respondents were informed about the research's purpose, their written consent was taken on questionnaire, and top management's permission was obtained before data collection. However, 523 responses were collected. In the preliminary analysis, eleven responses were eliminated due to missing data. Therefore, 512 responses are considered the sample size for this research, which is considered enough as recommended by [56].

### 3.3 Research instruments

This research has adapted measurement items from the existing studies to determine the findings. These items were selected to fit into the context of this research. The process of adaptation was based on a few modifications to change the context of the study. Furthermore, a panel of four reviewers was considered for the scale adoption process. They helped the researchers modify the scale items for research data collection. The panel of four reviewers confirmed the face and content validity of the adapted scale. However, the operationalization of this research variable is considered for scale items adapted to collect the data. The current study is based on primary data as the information from different manufacturing firms' employees is considered a research population. Empowering leadership is operationalized as providing support to the employees for their productive work. Scale items for this variable are adapted from the study of [57], for example, "I believe giving autonomy to employees improves their working". Furthermore, conflict management is operationalized as the approach to the employees for the management of conflicts between the employees, and the scale items for this variable are adapted from the study of [58], for example, "Conflicts are improved while having power to work". Furthermore, the employees' performance is operationalized as the measure of the performance of employees in the organization, and scale items for this variable are adapted from [59], for example, "I believe the performance of employees is improved with the support of management". Moreover, the organization's sustainability is operationalized as the performance of employees to improve the productive performance, and the scale items for this variable are adapted from S [60]), for example, "I acknowledge the management support for organizational sustainability". Finally, emotional stability is operationalized as the capability of the employees to control their emotions in any situation, and the measurements for this variable are adapted from [61], for example, "I feel confident about my working in the organization".

## 3.4 Data analysis method

This research has used Smart PLS 4 for analysis of data. The findings of measurement model assessment and structural model assessment were tested. The data was inserted into Smart PLS, and PLS Algorithm and PLS Bootstrapping were performed. This method is reliable for investigating the complex model of the research. This method was also employed in previous studies [62, 63]. The measurement model is tested to find individual items' reliability, convergent validity, discriminant validity, and standard method bias. The findings of the structural model are used to test the path findings. Similarly, the findings of coefficient of determination, effect size and predictive relevance are also checked in this research.

# 4. Data analysis and discussion

## 4.1 Normality of distribution

The normality of distribution is checked in any research to determine data validity. This research has considered the missing values, mean, mediation, and standard deviation findings to determine the data's normality. The normality of the distribution was achieved.

## 4.2 Measurement model assessment

The findings of measurement model assessment are used to check the findings of factor loadings. The findings of factor loadings are used to determine the reliability of individual items. The factor loading values should be greater than 0.60 [64] for significant items' validity. The items loaded below 0.60 should be deleted. The highlighted results in Table 1 confirmed that this study has achieved significant factor loadings. The findings of skewness and kurtosis are also taken after inserting the descriptive statistics data into Smart PLS 4. The skewness and kurtosis findings are tested with the thresholds +2 and -2 recommended by [65]. The skewness and kurtosis values between -2 and +2 are considered significant for the normality of the distribution.

The findings of Cronbach's alpha and composite reliability are determined to check the internal consistency of the research data. The findings of composite reliability should be greater than 0.70 [66], and the Cronbach alpha should be more than 0.70 [67] for significant internal consistency. Moreover, the variance between research data is checked with the findings of the average variance extracted. The findings of the average variance extracted should be more than 0.50 [68] for the significant variance of items loaded on constructs. The reported data in Table 2 confirmed internal collinearity between the research data.

The findings of discriminant validity are tested with measurement model assessments. It is tested to identify the possible multicollinearity issues in the research data. This research has used the Heterotrait-Monotrait (HTMT) method to determine the discriminant validity. According to [69], the findings in the HTMT table should be less than 0.90 for significant discriminant validity. The findings in Table 3 confirmed that the research data is appropriate and that there are no multicollinearity issues in this research.

Furthermore, the collinearity issues were tested with the findings of the variance inflation factor. This method confirms no collinearity issues when the findings of the variance inflation factor are less than 3.3 [70]. The reported data in Table 4 confirmed that the findings are significant and that the research data has no collinearity issues.

The effect size findings are identified to check the effect of one variable on another. The values of effect sizes 0.02, 0.15, and 0.35 are small, medium, and large effect sizes, respectively [71]. The effect of empowering leadership is medium on conflict management and employees'

**Table 1. Factor loadings and skewness & kurtosis.**

| Items | CM | EL | EP | ES | OS | Excess Kurtosis | Skewness |
|---|---|---|---|---|---|---|---|
| CM1 | 0.921 | | | | | -0.381 | 0.101 |
| CM2 | 0.92 | | | | | -0.536 | 0.47 |
| CM3 | 0.891 | | | | | -0.752 | 0.312 |
| CM4 | 0.922 | | | | | -0.783 | 0.392 |
| CM5 | 0.901 | | | | | -0.43 | 0.31 |
| EL1 | | 0.895 | | | | -0.672 | 0.235 |
| EL2 | | 0.909 | | | | -0.855 | 0.166 |
| EL3 | | 0.915 | | | | -0.747 | 0.203 |
| EL4 | | 0.896 | | | | -0.74 | 0.316 |
| EL5 | | 0.878 | | | | -0.733 | 0.367 |
| EP1 | | | 0.866 | | | -0.685 | 0.392 |
| EP2 | | | 0.881 | | | -0.603 | 0.365 |
| EP3 | | | 0.862 | | | -0.778 | 0.339 |
| EP4 | | | 0.783 | | | -0.429 | 0.455 |
| EP5 | | | 0.811 | | | -0.893 | 0.222 |
| ES1 | | | | 0.935 | | -0.623 | 0.32 |
| ES2 | | | | 0.915 | | -0.572 | 0.287 |
| ES3 | | | | 0.904 | | -0.095 | 0.606 |
| ES4 | | | | 0.901 | | 0.493 | 0.904 |
| OS1 | | | | | 0.905 | 0.839 | 0.938 |
| OS2 | | | | | 0.898 | 0.49 | 0.786 |
| OS3 | | | | | 0.904 | 0.552 | 0.682 |
| OS4 | | | | | 0.933 | 0.34 | 0.719 |
| OS5 | | | | | 0.939 | 0.491 | 0.83 |

EM = Empowering Leadership, CM = Conflict Management, ES = Emotional Stability, EP = Employees' Performance and OS = Organization Sustainability

performance. Furthermore, the effect of conflict management is small on organization sustainability, and the effect on employees' performance is significant.

The findings of the coefficient of determination are tested to check the variance between research data. The study has tested the variance. The reported results in Table 5 highlighted an 86% variation between conflict management and empowering leadership. Furthermore, the results highlighted an 81% variance in employees' training with empowering leadership. Finally, there is a 68% variance in achieving organization sustainability with conflict management and employee training.

The results of predictive relevance are checked to see if the dependent variance has the predictive ability of the independent variance. The findings of predictive relevance are also

**Table 2. Cronbach's alpha, composite reliability and average variance extracted.**

| Variables | Cronbach's Alpha | Composite Reliability | Average Variance Extracted (AVE) |
|---|---|---|---|
| CM | 0.949 | 0.961 | 0.83 |
| EL | 0.940 | 0.955 | 0.808 |
| EP | 0.896 | 0.924 | 0.708 |
| ES | 0.934 | 0.953 | 0.835 |
| OS | 0.952 | 0.963 | 0.839 |

EM = Empowering Leadership, CM = Conflict Management, ES = Emotional Stability, EP = Employees' Performance and OS = Organization Sustainability

**Table 3. HTMT.**

| Variables | CM | EL | EP | ES | OS |
|---|---|---|---|---|---|
| CM | | | | | |
| EL | 0.723 | | | | |
| EP | 0.883 | 0.784 | | | |
| ES | 0.754 | 0.721 | 0.753 | | |
| OS | 0.694 | 0.698 | 0.745 | 0.678 | |

EM = Empowering Leadership, CM = Conflict Management, ES = Emotional Stability, EP = Employees' Performance and OS = Organization Sustainability

highlighted in Table 5. The statistical results confirmed that empowering leadership has a predictive ability of 71% in conflict management [72]. Furthermore, the statistical results confirmed that employees' performance has a predictive ability of 57%. Finally, the statistical results confirmed that conflict management and employee performance have a predictive ability of 57% in conflict management.

## 4.3 Structural model assessment

The findings of the structural model assessment are checked to determine the research findings. The $t > 1.96$ is considered a significant threshold for a significant relationship because this research has non-directional hypotheses [73]. The findings of H1 reported a positive and significant relationship between empowering leadership and conflict management. The findings of H2 reported a positive and significant relationship between empowering leadership and employees' performance. The findings of H3 reported a positive and significant relationship between conflict management and organization sustainability. The findings of H4 reported a positive and significant relationship between employees' performance and organization sustainability. Meanwhile, H5 results reported a significant moderating role of emotional stability between empowering leadership and conflict management. Finally, H6 data confirmed a moderating role of emotional stability between empowering leadership and employees' performance. The results are reported in Table 6.

**4.3.1 Moderating hypotheses results.** There are two moderating relationships in this research. H5 is the first moderating effect, and the results reported a significant moderating role of emotional stability between empowering leadership and conflict management. The results highlight that increased emotional stability increases the relationship between empowering leadership and conflict management. Therefore, the change in empowering leadership in the presence of emotional stability positively influences conflict management. The findings are displayed in Fig 2.

**Table 4. Common method bias.**

| Variables | CM | EL | EP | ES | OS |
|---|---|---|---|---|---|
| CM | | | | | 2.801 |
| EL | 2.954 | | 2.954 | | |
| EP | | | | | 2.801 |
| ES | 2.954 | | 2.954 | | |
| OS | | | | | |

EM = Empowering Leadership, CM = Conflict Management, ES = Emotional Stability, EP = Employees' Performance and OS = Organization Sustainability

**Table 5. Coefficient of determination and predictive relevance.**

| Variables | R Square | R Square Adjusted | |
|---|---|---|---|
| CM | 0.863 | 0.862 | |
| EP | 0.817 | 0.815 | |
| OS | 0.685 | 0.683 | |
| Variables | SSO | SSE | $Q^2$ (= 1-SSE/SSO) |
| CM | 1125 | 326.235 | 0.710 |
| EL | 1125 | 1125 | |
| EP | 1125 | 476.96 | 0.576 |
| OS | 1125 | 484.131 | 0.570 |

EM = Empowering Leadership, CM = Conflict Management, EP = Employees' Performance and OS = Organization Sustainability

H6 is the second moderating effect, and the results reported a significant moderating role of emotional stability between empowering leadership and employees' performance. The results highlight that increased emotional stability increases the relationship between empowering leadership and employees' performance. Therefore, the change in empowering leadership in the presence of emotional stability positively influences employee performance. The findings are displayed in Fig 3.

## 4.4 Discussion

The current research has achieved its objectives by finding the answers to initially developed questions. The research findings are tested and checked, considering existing studies' theoretical and empirical findings. The current study highlighted that the effect of empowering leadership is significant on conflict management, and the first hypothesis is accepted. However, this relationship has substantial support from the existing studies in the literature. According to [31], if an employee is motivated, they need a plan to continue working for the company. [60] found that this element can affect how well employees perform when the strategic management strategy is applied to empower the workforce. Similarly, [58], when employees are adequately supported, team conflict management becomes effective. According to [42], these employees' disagreements are bad for their ability to execute consistently. The results of [52] reveal that employees are motivated to increase their influence over others to obtain work by using their leadership skills. Hence, the findings of this hypothesis are in line with the findings of previous studies and support the developed hypothesis.

**Table 6. Direct and indirect hypotheses.**

| Hypothesis | Paths | Original Sample (O) | Sample Mean (M) | Standard Deviation (STDEV) | T Statistics (|O/STDEV|) | P Values |
|---|---|---|---|---|---|---|
| | | | | Direct Paths | | |
| 1 | EL -> CM | 0.539 | 0.536 | 0.072 | 7.537 | 0.000 |
| 2 | EL -> EP | 0.473 | 0.474 | 0.074 | 6.428 | 0.000 |
| 3 | CM -> OS | -0.270 | -0.269 | 0.074 | 3.668 | 0.000 |
| 4 | EP -> OS | 1.048 | 1.047 | 0.060 | 17.384 | 0.000 |
| | | | | Moderating Paths | | |
| 5 | Moderating Effect 1 -> CM | 0.412 | 0.414 | 0.072 | 5.710 | 0.000 |
| 6 | Moderating Effect 2 -> EP | 0.429 | 0.428 | 0.072 | 5.995 | 0.000 |

EM = Empowering Leadership, CM = Conflict Management, ES = Emotional Stability, EP = Employees' Performance and OS = Organization Sustainability

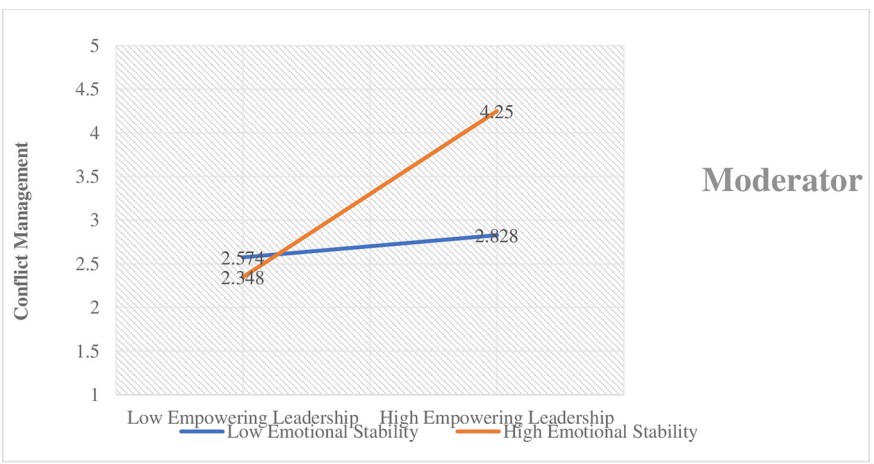

EM = Empowering Leadership, CM = Conflict Management, ES = Emotional Stability, EP = Employees' Performance and OS = Organization Sustainability

**Fig 2. Moderating effect 1.**

Furthermore, the current study emphasizes that the effect of empowering leadership is significant on employees' performance, and the second hypothesis is accepted. This relationship has substantial support from the existing studies in the literature. According to [59], any company's employees perform better when they have a solid working philosophy and feel that the organizational support is acceptable. According to [39], employee performance fluctuates throughout time, yet strategic working aids in achieving sustainability in performance. According to [33], when employees can communicate their creative ideas to management, any organization's operations can be enhanced. According to [41], businesses in the manufacturing industry must raise employee performance standards if they want to supply the market with uniform products. According to [16], the performance of the employees varies when the skills of any organization are set to fulfil the necessary criteria. The analysis of existing studies' findings provides rational support for the findings of this hypothesis.

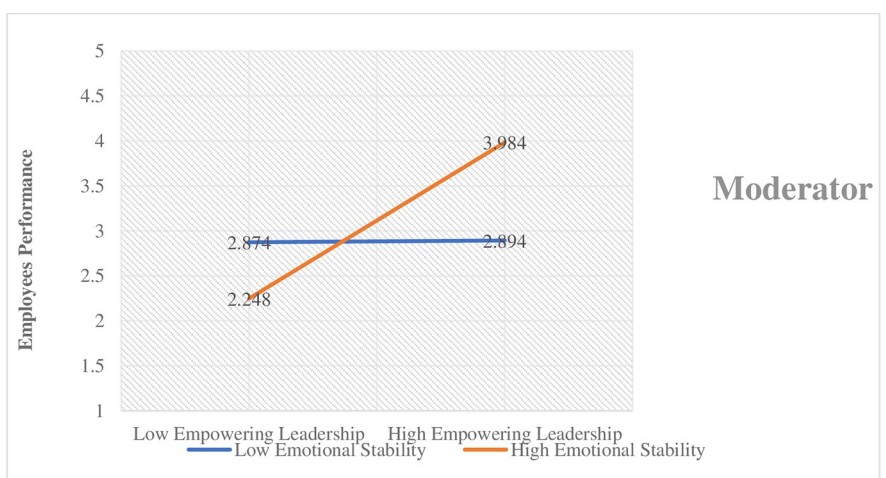

EM = Empowering Leadership, CM = Conflict Management, ES = Emotional Stability, EP = Employees' Performance and OS = Organization Sustainability

**Fig 3. Moderating effect 2.**

Thirdly, the current study established that the effect of conflict management is significant on organization stability, and the third hypothesis is accepted. However, this relationship has substantial support from the existing studies in the literature. According to [57], the efficiency of any business must be improved through teamwork because it is the most effective strategy to attain sustainability goals. According to [46], employees perform better and uphold organizational standards when management is responsible for giving them authority. According to [28], conflict management techniques can improve any organization's employee conflicts. According to [19], when people work in teams without tension, their innovative performance enables them to obtain better prospects. According to [22], to ensure that employees work productively, the management's primary obligation is to reduce employee conflict. The findings of these studies provided ground and supported this research results.

Fourthly, this study highlighted how employees' performance is significant in organization stability, and the fourth hypothesis is accepted. However, this relationship has substantial support from the existing studies in the literature. Similar results were found by [20], the results of the study reveal that employees with access to tools and training can function productively and essentially for their long-term success. Conferring to [54], the management must concentrate on the available resources and lessen employee conflict to improve employee direction and boost organizational performance. Giving to [47], businesses today demand that their staff perform productively. According to [50], when an organization has a competitive edge, it can remain viable. According to [74], teamwork is also necessary for employees to advance their tasks fruitfully. The empirical discussion of the existing studies supported the results of current research.

This research has determined that the moderating influence of emotional stability positively influences the relationship between empowering leadership and conflict management. The fifth hypothesis of this research is approved in this way, but there is theoretical support for this relationship by existing studies. According to [61], employees' emotional empowerment enables them to get past their problems and lessens their reliance on the organization's effectiveness. It was found by [36] that advanced behaviour is achievable when employees take steps to increase their productive behaviour at work. According to [45], people cannot separate themselves from their emotions because they are a natural component of existence. [44] reveal that employees may have more opportunities to work towards a competitive edge if they better understand what constitutes productive performance. Finally, the study of [37] indicated that the management strategy of employee empowerment can increase productivity since it enables workers to resolve issues and contribute to the company's long-term viability. The theoretical justifications and findings of this research support this moderating relationship.

Lastly, the current study established that the moderating influence of emotional stability positively influences the relationship between empowering leadership and employees' performance. The sixth hypothesis of this research is approved in this way, but there is theoretical support for this relationship by existing studies. According to [24]), the sustainable working method is made possible whenever the company's management adopts an empowering strategy for the employees. According to [34], it is necessary to enhance organizational performance and progress when employees of any organization are not given the proper support. According to [30], every employee undoubtedly has a unique set of values and a different way of working. Still, when the leadership is given the authority to enhance organizational performance, the organization's performance can become more productive. According to [35], as management assistance for employees increases their productive performance, a deliberate strategy to improve organizational performance is required. According to [26], although the management of many businesses uses various working methods, it is essential to ensure that employees are moving correctly.

## 5. Theoretical implication, practical implications, future directions

This research has theoretical importance as the findings of existing studies in the literature didn't determine the relationship established by it. This research confirmed the impact of empowering leadership on conflict management and employees' performance in the manufacturing sector. The loop in literature is closed based on these findings. Furthermore, this research also established that conflict management and employee performance are significant antecedents of organizational sustainability. The existing studies should be inconsistent between these relationships. However, another loop in the literature is also closed by the findings of this research. Furthermore, the study introduced moderating variables in the proposed framework. The study determined inconsistency in existing literature, but current research findings highlighted that the moderating influence of organization sustainability positively influences and strengthens the relationship between empowering leadership and conflict management. Furthermore, current research findings confirm that the moderating influence of organization sustainability has a positive influence and strengthens the relationship between empowering leadership and employees' performance.

The research has practical importance because its findings are reliable for the manufacturing sector. The study established that an organization's sustainability is possible through employees' performance and conflict management. Therefore, it is recommended that the top management of manufacturing firms have a transformative leadership approach to encourage middle management and employees to work with their innovative ideas. The research has established that innovative ideas are required to improve the performance of any firm. Still, firms should establish an empowering leadership approach to empower their employees for productive performance at work. Indeed, this research pointed out that when employees are motivated by an empowering leadership approach, they perform comparatively well. Moreover, this research asserts that emotional stability is necessary as it helps to strengthen the relationship between empowering leadership for productive conflict management and the reliable performance of employees. These practises implemented in the manufacturing sector can improve the performance of employees and lead to organisational sustainability.

### 5.1 Limitations and future directions

Although, this research has closed the loop in the literature with its significant findings. This study has limitations as it considers data from one country. However, the dynamics have changed in other countries and working environments. Secondly, this study has limitations as it tested the data with a structural equation model, but regression analysis was performed in the existing studies. Thirdly, this study has limitations as it hasn't tested the mediating role of employees' performance and conflict management. Yet, there are some recommendations for this research for future studies. Future studies must determine the antecedents that could influence the employees to empower the leadership approach, which would significantly contribute to the literature. Furthermore, future studies should test the moderating role of employees' psychological empowerment in the relationship between conflict management and organization sustainability. Moreover, this research has determined the findings based on the sample of middle-level management employees. Still, future studies should collect data from line workers to determine their observations for organizational performance and employee management. These future directions would improve the body of knowledge related to organization sustainability.

## Author Contributions

**Conceptualization:** Yi Wang.

**Data curation:** Yi Wang.

**Formal analysis:** Yi Wang.

**Methodology:** Yi Wang.

**Software:** Yi Wang.

**Validation:** Yi Wang.

**Writing – original draft:** Yi Wang.

**Writing – review & editing:** Yi Wang.

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
