## [Decision Letter · Decision Letter 0]

18 Sep 2023

PONE-D-23-25025Empowering Leadership: A Conflict Resolver and A Performance Booster for OrganizationsPLOS ONE

Dear Dr. Wang,

Thank you for submitting your manuscript to PLOS ONE. After careful consideration, we feel that it has merit but does not fully meet PLOS ONE’s publication criteria as it currently stands. Therefore, we invite you to submit a revised version of the manuscript that addresses the points raised during the review process.

We look forward to receiving your revised manuscript.

Kind regards,

Faisal Shafique Butt, Ph.D

Academic Editor

PLOS ONE

Journal Requirements:

Reviewers' comments:

Reviewer's Responses to Questions

**Comments to the Author**

1. Is the manuscript technically sound, and do the data support the conclusions?

Reviewer #1: Yes

Reviewer #2: Yes

Reviewer #3: Yes

2. Has the statistical analysis been performed appropriately and rigorously? 

Reviewer #1: Yes

Reviewer #2: No

Reviewer #3: Yes

3. Have the authors made all data underlying the findings in their manuscript fully available?

Reviewer #1: No

Reviewer #2: Yes

Reviewer #3: Yes

4. Is the manuscript presented in an intelligible fashion and written in standard English?

Reviewer #1: No

Reviewer #2: No

Reviewer #3: Yes

5. Review Comments to the Author

Reviewer #1: Dear Editor,

Thank you very much for giving me this opportunity to review the manuscript entitled “Empowering Leadership: A Conflict Resolver and A Performance Booster for Organizations ", I have a few suggestions for the author(s) to incorporate in the manuscript and improve its quality further.

1. Revise the abstract. it should not be in questionable form. The abstract is not well-written and explicitly stated.

2. Revise your Title.

3. Too many sentences are used that give the same meaning in the whole document which creates repetition.

4. Thoroughly review the hypotheses and make them relevant to the literature.

5. The Research instrument's definition is non-specific.

6. Explain clearly in your document how the scale is selected and why.

7. The multicollinearity issue is not addressed, what is the reason behind this?

8. Describe each heading 1st (e.g. Table 1: - describe or explain it 1st), similarly, revise all headings.

9. Add future directions, limitations, and implications.

10. No equation is added to the methodology. Add mathematical models.

11. Is the Robustness check? If checked then add the equation in the methodology.

12. Revise and separate the methodological section with equations from the results.

13. What is the rationale behind using underlying methods in the current research?

14. Why the authors did not employ other similar techniques and prefer this technique solely?

15. Are there any other studies on a similar methodology? If yes, please cite them to have a better empirical justification.

16. What are the future prospects of this study? Adding the future directions along with

the implications for the individual market would be the greater advantage of the study.

17. I found some errors in the references. In that case, a thorough proofreading of references is needed.

18. I found some grammatical and typo errors in the manuscript. In that case, a

professional proofreading can polish and enhance the quality of your manuscript.

Reviewer #2: I am thankful to the editorial office for allowing me to review this manuscript, titled “Empowering Leadership: A Conflict Resolver and A Performance Booster for Organizations”. The idea and topic of this study is attractive but there is a need of severe efforts for the improvement/betterment of the manuscript, so here given below are some suggestions/comments for the betterment/improvement of the manuscript:

1. It would be better if authors remove Q1, Q2 and Q3 from the abstract and write in plain form about the purpose of this study.

2. In abstract authors write down questions about mediating role of conflict management and employee performance instead of direct link of these variables with organization sustainability.

3. Special attention requires to check all references of the manuscript as many references show surname of the authors which need to remove.

4. Another major problem found in manuscript is about the grammatical mistakes, so it is suggested that authors must proofread the manuscript from native language proofreader.

5. Justification and argumentation about empowering leadership in relation with organization sustainability, conflict management and employee performance are missing, which need to be added in introduction section.

6. Before the last paragraph where questions of study are addressed, there is a need to add questions about mediating role of conflict management and employee performance.

7. There is a need for correction of word in heading “2.1 Proposed” instead of “purposed”.

8. The explanation of relationships of proposed model using theoretical lens of transformational leadership theory is unsatisfactory, it is suggested that authors explain all the relationships of the model using lens of overarching theory.

9. There is also needed to add more latest studies (2018-2023) in literature review section for the support of justifications of the study.

10. It is suggested that authors modify the hypotheses as given bellow format:

H1 There is a positive relationship between empowering leadership and organization sustainability.

H2 There is a positive relationship between empowering leadership and conflict management.

H3 There is a positive relationship between empowering leadership and employees’ performance.

H4 There is a positive relationship between conflict management and organization sustainability.

H5 There is a positive relationship between employees’ performance and organization sustainability.

H6 Conflict management mediates the relationship between empowering leadership and organization sustainability.

H7 Employee performance mediates the relationship between empowering leadership and organization sustainability.

H8 Emotional stability moderates the relationship between empowering leadership and conflict management in such sense that higher/lower level of emotional stability will strengthen/weaken this positive relationship.

H9 Emotional stability moderates the relationship between empowering leadership and employee performance in such sense that higher/lower level of emotional stability will strengthen/weaken this positive relationship.

11. In methodology section, authors first provide information with this sequence for better understanding of the readers:

3.1 Sampling frame and technique

3.2 Data collection and sample size

3.3 Research Instruments

3.4 Data analysis method

12. It is suggested that authors must provide a minimum of one sample item of each scale.

13. It is suggested that authors remove table 1 as there is no need of missing values, mean, median, min, max of each item.

14. It is suggested that authors add skewness and kurtosis values of the items in table 2 with factor loadings.

15. As authors used Smart-PLS 4 and provides HTMT values but Fornell-Larker values are missing which need to be added in HTMT table.

16. It is suggested that authors merge Table 8 and Table 9 into one table and place this table just after HTMT table.

17. It is suggested that authors remove table 7 (effect size).

18. It would be better for the readers if authors provide information first about direct effects between independent and dependent variables as no one research model can be formulated without formation of these relationships. So, in table 6 authors provide values in this sequence:

1. EL → OS

2. EL → CM

3. EL → EP

4. CM → OS

5. EP → OS

6. EL → CM → OS

7. EL → EP → OS

8. EL x ES → CM

9. EL x ES → EP

19. It is also suggested that authors must provide the values in three decimals instead of one (e.g., 0.000).

20. The interpretation of results especially of direct, indirect, and moderating results and interpretation of moderation graph is much weaker, therefore, it is suggested that authors must read the latest papers of this journal for impressive interpretation of the results.

21. It is suggested that authors rewrite the discussion section after formulation of mediation hypotheses of both mediators.

22. In practical implications section, this sentence is out of sense “Therefore, the top management of manufacturing firms must work on transformative leadership theory to encourage middle management and employees to work with their innovative ideas” why management work on theory, practical implications represent the suggestions by the authors based on the findings of the study, so it is suggested that authors must revisit this section.

23. The limitations section of this manuscript is missing, which needs to be added.

Reviewer #3: The paper could benefit from a more in-depth explanation of the theoretical framework, specifically the transformational leadership theory. While the paper references this theory, it lacks a clear and detailed exposition of its key principles, which is essential for readers to fully grasp the context of the study.

The paper relies on a relatively small sample size (512 respondents) from the manufacturing sector firms in Wuhan, China. Given the vast diversity in organizational contexts, industries, and cultures, the study's findings may not be generalizable beyond this specific context. A broader and more diverse sample would enhance the paper's external validity.

While the paper mentions that measurement items were adapted from existing studies, it does not provide sufficient details about the adaptation process or the validation of these items in the new context. A more robust explanation of how these items were selected and modified for the study is necessary for transparency and rigor.

The paper briefly mentions the use of Smart PLS 4 for data analysis but lacks a comprehensive discussion of the analytical procedures. Readers would benefit from a more detailed explanation of how the data were analyzed, including model specifications, estimation techniques, and potential limitations of the chosen approach.

While the paper appropriately cites existing literature, it occasionally relies on these references without critical engagement. A more critical analysis and synthesis of prior research, including discussions of conflicting findings or gaps in the literature, would strengthen the paper's contribution.

The discussion section is somewhat superficial in its treatment of the findings. It tends to summarize the results without delving into the practical implications of the research or providing nuanced interpretations. A more thorough discussion that explores the "why" behind the relationships observed and how they align with prior theories would add depth to the paper.

By addressing these weaknesses would significantly strengthen the paper and increase its suitability for publication.

6. PLOS authors have the option to publish the peer review history of their article (what does this mean?). If published, this will include your full peer review and any attached files.

Reviewer #1: **Yes: **RAMEEZA ANDLEEB

Reviewer #2: **Yes: **Muhammad Salman Chughtai, Faculty of Management Science, International Islamic University, Islamabad, Pakistan

Reviewer #3: **Yes: **Dr Zartashia Hameed

---

## [Author Response · Author response to Decision Letter 0]

17 Oct 2023

Dear Editor,

I am pleased to submit the revised manuscript, PONE-D-23-25025, entitled “Empowering Leadership: A Conflict Resolver and A Performance Booster for Organizations”. All three reviewers recommended minor revisions. All the comments of worthy reviewers are addressed, and the quality of the paper has improved substantially. Thanks for your valuable comments and insights and for helping me improve the quality of my paper. I hope that the revised version will meet the quality standard of PLOS One for publication in due course. Red and blue highlighted text indicates the changes made in manuscript. Thanks

Reviewer #1:

Thank you very much for giving me this opportunity to review the manuscript entitled “Empowering Leadership: A Conflict Resolver and A Performance Booster for Organizations ", I have a few suggestions for the author(s) to incorporate in the manuscript and improve its quality further.

Comment 1:

Revise the abstract. it should not be in questionable form. The abstract is not well-written and explicitly stated.

Response:

Dear reviewer, the abstract of this research is completely revised. Red text indicates the changes made in article.

Comment 2:

Revise your Title.

Response:

Respected reviewer, we believe that the title of study is already clear and concise. 

Comment 3:

Too many sentences are used that give the same meaning in the whole document which creates repetition.

Response:

Dear reviewer, we performed the professional proof reading of article to avoid such repetitions. Thanks for your valuable comments.

Comment 4:

Thoroughly review the hypotheses and make them relevant to the literature.

Response:

Dear reviewer, hypothesis is now improved as per your recommendation. Thanks for your valuable feedback.

Comment 5:

The Research instrument's definition is non-specific.

Response:

Dear reviewer, the research instruments operationalization in now revised to make it specific.

Comment 6:

Explain clearly in your document how the scale is selected and why.

Response:

Dear reviewer, the rationale for selection of scale items is explained in the revised methodology. Thanks for your comments.

Comment 7:

The multicollinearity issue is not addressed, what is the reason behind this?

Response:

Dear reviewer, the multicollinearity issues are addressed in HTMT findings by checking discriminant validity.

Comment 8:

Describe each heading 1st (e.g. Table 1: - describe or explain it 1st), similarly, revise all headings.

Response:

Dear reviewer, each heading is described clearly according to the content of the study. Please find the incorporated changes in manuscript.

Comment 9:

Add future directions, limitations, and implications.

Response:

Dear reviewer, the implications, limitations, and future directions are added. Changes can be seen in revised article through red highlighted text.

Comment 10:

No equation is added to the methodology. Add mathematical models.

Response:

Dear reviewer, Smart PLS is used for data analysis. No equations are required for performing analysis in Smart Pls.

Comment 11:

Is the Robustness check? If checked then add the equation in the methodology.

Response:

Dear reviewer, measurement model assessment and structural model assessment were performed to ensure the robustness of data.

Comment 12:

Revise and separate the methodological section with equations from the results.

Response:

Dear reviewer, the methodological section is separate from the data analysis section.

Comment 13:

What is the rationale behind using underlying methods in the current research? Why the authors did not employ other similar techniques and prefer this technique solely? Are there any other studies on a similar methodology? If yes, please cite them to have a better empirical justification.

Response:

Dear reviewer, the study is quantitative in nature. Therefore, we used the methodology keeping in view the nature of study. Also, references are given in methodology for using specific methods. Smart Pls is a latest widely used tool for data analysis.

Comment 14:

What are the future prospects of this study? Adding the future directions along with

the implications for the individual market would be the greater advantage of the study.

Response:

Dear reviewer, the future directions of the study are highlighted along with theoretical and practical implications based on its findings.

Comment 15:

I found some errors in the references. In that case, a thorough proofreading of references is needed.

Response:

Dear reviewer, we cited the reference by using end note software. We cross checked the references again as per your recommendation. Thanks for your valuable feedback.

Comment 18:

I found some grammatical and typo errors in the manuscript. In that case, a

professional proofreading can polish and enhance the quality of your manuscript.

Response:

Dear reviewer, the manuscript is now proofread by a professional native English speaker to ensure quality of writing. 

Reviewer #2: 

I am thankful to the editorial office for allowing me to review this manuscript, titled “Empowering Leadership: A Conflict Resolver and A Performance Booster for Organizations”. The idea and topic of this study is attractive but there is a need of severe efforts for the improvement/betterment of the manuscript, so here given below are some suggestions/comments for the betterment/improvement of the manuscript:

Comment 1:

It would be better if authors remove Q1, Q2 and Q3 from the abstract and write in plain form about the purpose of this study.

Response:

Dear reviewer, the abstract section is revised as per your recommendations.

Comment 2:

In abstract authors write down questions about mediating role of conflict management and employee performance instead of direct link of these variables with organization sustainability.

Response:

Dear reviewer, we are not considering the mediation analysis in our study. Therefore, we have not developed hypotheses and findings for mediation testing. We used the process approach rather than mediation. Thank you for valuable comments.

Comment 3:

Special attention requires to check all references of the manuscript as many references show surname of the authors which need to remove.

Response:

Dear reviewer, all the references are checked, and no citation is made manually. All citations are made by using citation software end note.

Comment 4:

Another major problem found in manuscript is about the grammatical mistakes, so it is suggested that authors must proofread the manuscript from native language proofreader.

Response:

Dear reviewer, the manuscript is undergone for proofreading. The grammatical mistakes are resolved. Blue text indicates the proof reading done in article.

Comment 5:

Justification and argumentation about empowering leadership in relation with organization sustainability, conflict management and employee performance are missing, which need to be added in introduction section.

Response:

Dear reviewer, as per your recommendations, the justification for these relationships is now incorporated with greater explanation in introduction section.

Comment 6:

Before the last paragraph where questions of study are addressed, there is a need to add questions about mediating role of conflict management and employee performance.

Response:

Dear reviewer, we do not conceptualize the mediation. This can be done by future studies. 

Comment 7:

There is a need for correction of word in heading “2.1 Proposed” instead of “purposed”.

Response:

Dear reviewer, the heading is corrected as per your recommendations. 

Comment 8:

The explanation of relationships of proposed model using theoretical lens of transformational leadership theory is unsatisfactory, it is suggested that authors explain all the relationships of the model using lens of overarching theory.

Response:

Dear reviewer, more elaboration of theory is now added for conceptualization of relationships. Thanks for your valuable feedback.

Comment 9:

There is also needed to add more latest studies (2018-2023) in literature review section for the support of justifications of the study.

Response:

Dear reviewer, as per your directions, the latest references are incorporated in the literature review section.

Comment 10:

It is suggested that authors modify the hypotheses as given bellow format:

H1 There is a positive relationship between empowering leadership and organization sustainability.

H2 There is a positive relationship between empowering leadership and conflict management.

H3 There is a positive relationship between empowering leadership and employees’ performance.

H4 There is a positive relationship between conflict management and organization sustainability.

H5 There is a positive relationship between employees’ performance and organization sustainability.

H6 Conflict management mediates the relationship between empowering leadership and organization sustainability.

H7 Employee performance mediates the relationship between empowering leadership and organization sustainability.

H8 Emotional stability moderates the relationship between empowering leadership and conflict management in such sense that higher/lower level of emotional stability will strengthen/weaken this positive relationship.

H9 Emotional stability moderates the relationship between empowering leadership and employee performance in such sense that higher/lower level of emotional stability will strengthen/weaken this positive relationship.

Response:

Dear reviewer, we did not conceptualize the mediation. We tested the process model to see how empowering leaders resolve conflicts and improve employee performance, and then this low conflict and high employee performance contribute to organizational sustainability.

Comment 11:

In methodology section, authors first provide information with this sequence for better understanding of the readers:

3.1 Sampling frame and technique

3.2 Data collection and sample size

3.3 Research Instruments

3.4 Data analysis method

Response:

Dear reviewer, this section is modified as per your recommendations. Thanks for your critical comments.

Comment 12:

It is suggested that authors must provide a minimum of one sample item of each scale.

Response:

Dear reviewer, as per your recommendations, a minimum of one sample is provided.

Comment 13:

It is suggested that authors remove table 1 as there is no need of missing values, mean, median, min, max of each item.

Response:

Dear reviewer, as per your directions, the said table is removed.

Comment 14:

It is suggested that authors add skewness and kurtosis values of the items in table 2 with factor loadings.

Response:

Dear reviewer, as per your directions, the findings of skewness and kurtosis are added with factor loadings.

Comment 15:

As authors used Smart-PLS 4 and provides HTMT values, but Fornell-Larker values are missing which need to be added in HTMT table.

Response:

Dear reviewer, the HTMT is appropriate methodology for determining discriminant validity or multicollinearity issues. 

Comment 16:

It is suggested that authors merge Table 8 and Table 9 into one table and place this table just after HTMT table.

Response:

Dear reviewer, the said changes are made in the revised manuscript. 

Comment 17:

It is suggested that authors remove table 7 (effect size).

Response:

Dear reviewer, the table of effect size is removed as per your recommendations.

Comment 18:

It would be better for the readers if authors provide information first about direct effects between independent and dependent variables as no one research model can be formulated without formation of these relationships. So, in table 6 authors provide values in this sequence:

1. EL → OS

2. EL → CM

3. EL → EP

4. CM → OS

5. EP → OS

6. EL → CM → OS

7. EL → EP → OS

8. EL x ES → CM

9. EL x ES → EP

Response:

Dear reviewer, thanks for your comments . However, the relationships we tested in our study are already reported in the required table. We didn’t consider the mediating relationships. We just tested direct and moderating relationships.

Comment 19:

It is also suggested that authors must provide the values in three decimals instead of one (e.g., 0.000).

Response:

Dear reviewer, the values in three decimals are provided in the study.

Comment 20:

The interpretation of results especially of direct, indirect, and moderating results and interpretation of moderation graph is much weaker, therefore, it is suggested that authors must read the latest papers of this journal for impressive interpretation of the results.

Response:

Dear reviewer, the discussion is improved as per your recommendations. Please see the changes in discussion section.

Comment 21:

It is suggested that authors rewrite the discussion section after formulation of mediation hypotheses of both mediators.

Response:

Dear reviewer, we have not considered mediation analysis in this study. Thanks for your concerns.

Comment 22:

In practical implications section, this sentence is out of sense “Therefore, the top management of manufacturing firms must work on transformative leadership theory to encourage middle management and employees to work with their innovative ideas” why management work on theory, practical implications represent the suggestions by the authors based on the findings of the study, so it is suggested that authors must revisit this section.

Response:

Dear reviewer, as per your recommendations, the said changes are made accordingly. Please find the changes in implications section.

Comment 23:

The limitations section of this manuscript is missing, which needs to be added.

Response:

Dear reviewer, the limitations of the study are added in the revised manuscript.

Reviewer #3: 

Comment 1:

The paper could benefit from a more in-depth explanation of the theoretical framework, specifically the transformational leadership theory. While the paper references this theory, it lacks a clear and detailed exposition of its key principles, which is essential for readers to fully grasp the context of the study.

Response:

Dear reviewer, as per your recommendation the more elaboration of theoretical support is added. Thanks for your valuable feedback.

Comment 2:

The paper relies on a relatively small sample size (512 respondents) from the manufacturing sector firms in Wuhan, China. Given the vast diversity in organizational contexts, industries, and cultures, the study's findings may not be generalizable beyond this specific context. A broader and more diverse sample would enhance the paper's external validity.

Response:

Dear reviewer, the sample size is appropriate as recommended by hair et al.

Comment 3:

While the paper mentions that measurement items were adapted from existing studies, it does not provide sufficient details about the adaptation process or the validation of these items in the new context. A more robust explanation of how these items were selected and modified for the study is necessary for transparency and rigor.

Response:

Dear reviewer, the information about the adaption of measurement is now explained with more depth for improving the understanding of the readers.

Comment 4:

The paper briefly mentions the use of Smart PLS 4 for data analysis but lacks a comprehensive discussion of the analytical procedures. Readers would benefit from a more detailed explanation of how the data were analyzed, including model specifications, estimation techniques, and potential limitations of the chosen approach.

Response:

Dear reviewer, further information about sample is provided regarding the use of Smart PLS. 

Comment 5:

While the paper appropriately cites existing literature, it occasionally relies on these references without critical engagement. A more critical analysis and synthesis of prior research, including discussions of conflicting findings or gaps in the literature, would strengthen the paper's contribution.

Response:

Dear reviewer, the discussion section is now improved as per your recommendation. Thanks for your valuable feedback.

Comment 6:

The discussion section is somewhat superficial in its treatment of the findings. It tends to summarize the results without delving into the practical implications of the research or providing nuanced interpretations. A more thorough discussion that explores the "why" behind the relationships observed and how they align with prior theories would add depth to the paper. By addressing these weaknesses would significantly strengthen the paper and increase its suitability for publication.

Response:

Dear reviewer, the discussion section of the study is revised in the manuscript. The changes can be seen in red highlighted text.

---

## [Editor Report · Decision Letter 1]

31 Oct 2023

Empowering Leadership: A Conflict Resolver and A Performance Booster for Organizations

PONE-D-23-25025R1

Dear Dr. Wang,

We’re pleased to inform you that your manuscript has been judged scientifically suitable for publication and will be formally accepted for publication once it meets all outstanding technical requirements.

Kind regards,

Faisal Shafique Butt, Ph.D

Academic Editor

PLOS ONE
---

## [Editor Report · Acceptance letter]

7 Nov 2023

PONE-D-23-25025R1 

Empowering Leadership: A Conflict Resolver and A Performance Booster for Organizations 

Dear Dr. Wang:

I'm pleased to inform you that your manuscript has been deemed suitable for publication in PLOS ONE. Congratulations! Your manuscript is now with our production department. 

Kind regards, 

on behalf of

Dr. Faisal Shafique Butt 

Academic Editor

PLOS ONE